# A Survey on Methodological Issues of Clinical Research Studies Reviewed by Independent Ethic Committees during the COVID-19 Pandemic in Italy

**DOI:** 10.3390/ijerph191811673

**Published:** 2022-09-16

**Authors:** Alberto Milanese, Paolo Trerotoli, Annarita Vestri

**Affiliations:** 1Department of Public Health and Infectious Diseases, Sapienza University of Rome, 00185 Rome, Italy; 2Department of Biomedical Science and Human Oncology, University of Bari Aldo Moro, 70121 Bari, Italy

**Keywords:** COVID-19 pandemic, independent ethics committees, research, surveys and questionnaire

## Abstract

The struggle for information and the hasty search for answers caused by the COVID-19 pandemic threatened the possibility of lowering study quality, as well as ethical committees’ review standards during the outbreak. Our investigation aimed to assess the impact of COVID-19 on the quality of clinical research studies submitted to Italian Ethics Committees in the period between April and July 2020. All 91 Italian ethics committees were contacted via email in order to collect anonymized information on the type and quality of COVID-19-related studies submitted to each committee during the study period. The present study summarizes the characteristics of the 184 study applications collected, pointing out, especially, how the quality of the study population and statistical analysis are crucial variables in determining the study approval. Nevertheless, despite the need for high-quality and open scientific information, especially exacerbated by this particular historical period, only a minority of the ethics committees (20.9%) agreed to share their data; such scarce participation, beyond biasing the representativeness of the results obtained by the present study, more importantly, hinders the broader goal of creating trust between researchers and the general public.

## 1. Introduction

Since its first identification in December 2019, the coronavirus disease 2019 (COVID-19) pandemic has spread from the city of Wuhan in China all over the world. The virus was first confirmed to have spread to Italy on 31 January 2020, and since then, as of April 2021, Italy became the eighth country in the world in terms of the absolute count of confirmed cases. Since the beginning of the pandemic, scientists and physicians from all over the world, Italy included, have been conducting many clinical studies involving affected patients in order to overcome the lack of information necessary in the battle against the novel virus. For example, more than 23,000 papers were indexed on major scientific databases such as Web of Science and Scopus [1].

The problem of methodology in research was highlighted many times as the number of research papers increased because of the increasing interest in the publishing of medical research by physicians [2]. Moreover, the spread of the novel disease created the need for new research, which needed research topics and a global coordination effort to be prioritized, as suggested by WHO roadmaps [3,4] in several areas such as natural history, prevention and control measures, transmission mechanisms, diagnostics, therapies, and vaccines. These areas generated the need for both observational and experimental studies to determine public health measures and clinical action to address the virus-related diffusion, morbidity, and mortality. In specific and particular contexts, the fast procedures implemented helped to focus on the main protocols that could have a true value for the patients, as was the case in the RECOVERY trial [5], while in others, the creation of a fast track did not help clinical research, as the authors declared that the newly implemented ad hoc committees’ system quickly collapsed, causing a paradoxical barrier to local research being performed, due to lengthy review times [6].

The haste caused by the global situation and the struggle for information threatened lowering study quality as well as ethical committees’ review standards during the outbreak [7,8]. The need for a new approach to ethics review during emergencies was described in previous scenarios such as the influenza pandemic (H1N1) or in previous coronaviruses’ spread (SARS and MERS).

Researchers felt delays and missed opportunities related to the evaluation of protocols, and therefore speed and flexibility, without losing the sight of harm and benefit for patients [9]. Even during the Ebola outbreak, the pressure for swift approval by the ethics committee was felt as a problem due to the urgent need for important new information relating to epidemiology and treatment. Several issues emerged in the WHO commission, and among these, the study design was considered, because the scientific solidity of a protocol is related to its social value, i.e., the benefit for each patient [10]. In other countries, surveys have been conducted on ongoing research and on the work of ethics committees to evaluate such problems [6,11].

Independent ethics committees (IECs) have a fundamental role in promoting the values of research ethics and in ensuring good clinical practice and valid methodology, therefore granting the quality and validity of the conclusions derived from clinical studies. Ethics committees must pay the utmost attention to methodological quality, especially in evaluating projects involving observational objectives, characterized by small sample sizes, and without clear hypotheses to verify. The need to quickly identify useful clinical or research ideas can justify accelerated data collection on limited numbers; however, information collection without appropriate planning can affect study results, pointing to inappropriate choices with a significantly negative clinical impact. In emergency situations, ethics committees need to optimize their procedures and establish an appropriate and flexible mechanism to avoid the ethical review and evaluation process, leading to delays in the start of research that is useful in responding to the health emergency.

In reports dedicated to the ethics of research in emergency situations, the World Health Organization stresses that, while research involving significant risks to the individuals or to the populations involved always requires full evaluation, for other types of research, it is conceivable to adopt a “fast track” approach. Protocols that involve no more than minimal risk and burden for the participants can be reviewed on an accelerated basis by one or more members rather than the entire committee if the ethics committee has established written procedures that allow this procedure. The ethics committees’ evaluation activity must always consider the objectives of a research, the methods and the design of the study, an adequate assessment of risks and benefits for the participants, the completeness and comprehensibility of information to the patient, and the involvement or otherwise of vulnerable subjects. Other actions proposed by the WHO to reduce the review times of emergency research projects include the increase in the frequency of communications between the components of the ethics committee and the incremental use of technology and electronic systems, together with the preparation of standard models of protocols to be modified by the researchers in the drafting of their research protocol in order to be submitted to the ethics committees for evaluation [12].

Our investigation aimed to assess the impact of COVID-19 on clinical research studies quality submitted to Italian Ethics Committees in the period between April and July 2020 and how the emergency could have influenced IECs’ work.

## 2. Materials and Methods

All 91 Italian ethics committees were contacted via email in order to collect information on the type and quality of COVID-19-related studies submitted to each committee in the period between April and July 2020. Among the 91 committees, 19 agreed to participate in the study (4 of them specified that they reviewed no COVID-19-related studies), 64 did not answer the survey, and 8 were not reachable despite repeated telephone and email attempts at contact. In summary, 15 ethical committees from 7 different Italian regions agreed to participate in the study and contributed with at least one case of COVID-19-related study, and an overall total of 184 study applications were collected. Additional information on the Italian ethics committees’ participation in the study is shown in Table 1. The committees were requested to fill in the information of the COVID-19-related reviewed studies in a spreadsheet file containing drop-down predefined options for any given column. The defined variables were anonymous study ID, study design, mono- or multicentric organization, choice of the comparison group, method of groups creation, study population, study objective and outcome, presence of sample size calculation, appropriateness concerning the study objective, description of statistical analysis, appropriateness for the study outcome, and the ethics committee’s final decision on the study application.

Results were summarized as counts and percentages and comparison among independent groups were performed using a chi-square test or Fisher’s exact test as appropriate. Univariate odds ratios (ORs) and their exact confidence intervals were determined to describe, for the main features of the studies described above, the chance of approval or rejection/referral-for-modification. The analyses were performed by SAS 9.4 for PC (SAS Institute Inc., 2016, Cary, NC, USA), and *p*-values < 0.05 were considered for statistical significance.

## 3. Results

The vast majority of 184 study applications included observational studies of various kinds (n = 164; 89.2%) compared to 19 experimental studies, accounting for only 10.3% of the total; moreover, a meaningful percentage (39.5%) of the reviewed applications were part of multicentric studies. The main study population was shown to be adults (n = 145; 78.8%) followed by pediatric (12.0%) and healthcare professionals (6.5%). A total of 139 studies (75.5%) were designed as single-arm experiments, and thus the results obtained were mainly determined without a comparison group. Furthermore, randomization was rarely used as a comparison group-creation method (2.7%), and in general, information on this specific topic was quite unclear, especially because it was biased by a consistently high percentage of missing answers (18.5%). The information relevant to the study objectives and outcomes, probably due to its intrinsic variability, was shown to be difficult to fit into a few predefined categories, as proven by the high percentages of “Other” responses collected (42.9% and 28.8%, respectively). Regarding the study quality information, we requested that each ethics committee declare if statistical analysis and sample size estimations were available and establish if the aforementioned were appropriate, relative to the single study’s objective and outcome. The description of the statistical analysis was largely included in the applications (77.7% present; 73.4% present and appropriate), while the sample size calculation was in contrast very low (29.9% present; 25% present and appropriate). In summary, only 71 studies (22.4%) presented an appropriate description of the statistical analysis and a correct sample size determination. The vast majority of the studies were approved (n = 127; 69.0%), while 34 (18.5%) were referred for modification and 6 (3.3%) were rejected; we highlight that for 17 studies, this information was missing but can be traced back to the overall contribution of a single ethics committee. All the reviewed study characteristics are shown in detail in Table 2 and Table 3.

Focusing on methodology, we have observed that IECs could have examined the appropriate methodology in 75.5% (117/155) of observational studies and 63.2% (12/19) of experimental studies, and the difference was not statistically significant (chi-square = 1.34, DF = 1, *p* = 0.2468). Sample size determination was applied in 24.7% (38/154, one missing value for sample size determination in the observational group) of observational studies and in 57.9% (11/19) of experimental studies, and the difference was statistically significant (chi-square = 9.19, DF = 1, *p* = 0.0024).

As an additional consideration, we tried to analyze whether approval, rejection, or referral for modification—conditions proving that the review efforts of the IEC aimed at study quality—were influenced by any of the variables collected (Figure 1).

Using Fisher’s exact test, we found a statistically significant association between application approval and study population, with pediatric population-based studies more frequently rejected or suspended in comparison with non-pediatric ones (OR = 0.2, IC 95%: 0.1, 0.6; *p*-value = 0.001), probably due to the greater caution generally related with pediatric healthcare, and between application approval and accurate description of statistical analysis, where studies reporting a good-quality statistical analysis were more likely to obtain IEC approval (OR = 11.0, IC 95%: 4.5, 27.9; *p*-value < 0.001).

## 4. Discussion

The main aim of this survey was to evaluate the effort of ethical boards to examine protocols, given the increased number of experimental and observational research about the SARS-CoV-2 virus and its related disease. Across the world, an increase was observed in research, and it turned into an attempt to shorten the time of protocol review through special procedures or separate review boards. These were necessities felt by many researchers and organizations, mostly because doctors and populations needed to face a new disease that was unknown in many ways, for example, with respect to risk factors, prognostic factors, prevention strategies, diagnostic accuracy, and therapeutics. What happened was a change in the need for evidence: before the pandemic, doctors and health workers, at every level, wanted good research with a higher level of trustworthiness; on the contrary, in the pandemic, and at this moment as well, we all need fast evidence [13].

The first element that needs criticism in our survey was the low participation rate of boards. Some boards manifested their reluctance to participate in the survey for privacy and confidentiality for research and researchers, and other committees did not participate because they were pressed by the higher number of protocols to evaluate, facing the attempt to be prompt for their duty rather than to give data for a survey. A survey on members of ethical committees for the evaluation of vaccine protocol on a human subject was performed as well, and the participation rate was about 62% [11]; the authors declare that a more general survey was necessary for better generalizability of results, but our experience showed the difficulties of having large participation in an emergency condition.

The results of our survey have mainly shown a great number of observational studies on COVID-19. This result could be read as the necessity to have as much information as possible about different issues: in a previously cited editorial in *Nature*, it was highlighted that in epidemiology, randomized studies sometimes may not be feasible [13], and therefore, observational studies could be the best researchers could do. Many authors have highlighted that in an emergency context, the promptness is valued with respect to rigor, but without losing quality and transparency [14]; furthermore, the urgency for the discovery should not lead to low-quality research [15]. We should recall that observational studies are in a lower-ranked position with respect to clinical trials in producing evidence-based medicine. It should be underlined that in this survey, we found no difference in appropriateness of methodology between experimental studies and observational ones, so researchers have presented suitable proposal, and IECs selected wisely and approved.

We observed a small number of comparative studies, most of all because of a lack of participation, and probably because of the special procedure for approval of clinical trials related to COVID-19, established in Italy. All experimental protocols have been directed to a single National Ethical Committee, to obtain faster approval, since the end of April 2020. This was an operative choice in accordance with procedures implemented everywhere in the world. In the same period, the approval for the clinical trial regarding Hydroxicloroquine and Dexamethasone were fast-tracked without loss of quality in the evaluation process, nor in the conduction of the trial, in terms of reduction in bureaucracy, integrated and smart data collection, trust in researchers and in the resulting value for patients and society, transparency, quick dissemination of results, and publicly available protocol. These were the key points that resulted in valuable results and the appreciation of procedures [5].

In this emergency, another issue could affect comparative studies: the choice of an appropriate control group and the application of randomization. Even if the higher number of cases could be a source of participants in the studies, it could be difficult to adjust for age, comorbidities, and duration of the illness. Therefore, all of these factors could affect the evaluation of the efficacy of interventions and could have a lower impact in the wider use of results [16]. Randomization could be troublesome because, in this emergency, a conflict between individual interest and social utility could arise [16]. We should not forget that researchers have aimed to reduce uncertainty, but the risk of having trials that are uncontrolled or without randomization, which give false-positive results, is too high and could not be justifiable with the emergency status [15].

An issue that emerged from our survey was the plan of the study and the determination of sample size, often disregarded in observational protocols, especially in retrospective design and/or in cross-sectional studies. In the experience of the authors, as a component of IECs, this happens not only during an emergency period. The correct planning is a basic value for any clinical study, and it is well known that sample size plays an important role in the importance and generalizability of results. A clinical study should result in social value: a benefit for patients and health workers should be achievable. Sample size, in any case, depends on the expected outcome, and in epidemics, it was difficult to define an expected outcome. Therefore, studies could be underpowered. Researchers could realize this problem and decide to extend the duration of the study and the enrollment [15,16]. At the same time, in the database of COVID-19 trials, 40% had a sample size smaller than 100 patients, a number that, if not determined under the right assumption, could be insufficient for achieving reliable results [13]. Solutions could come from multicenter studies that allow more patients to be recruited in less time, even if obstacles to sharing data and regulation among countries are overcome [4,15]. Furthermore, conducting multicenter studies could prevent the repetition of trials, which was a problem during the pandemic, because there were more trials without solid implants, and this phenomenon could affect the next step, the production of reliable reviews [13].

The speed and volume of data published in recent years on COVID-19 have made clear the need, which has been recognized for some time, to protect the integrity of scientific research. Editors and authors of scientific journals should be proactive in ensuring that the review status of their publications is clear and explicit: journals should declare whether and how an article was peer-reviewed. All preprint platforms should, as a standard policy, report to the reader that studies have not been peer-reviewed and should therefore be read with due caution. Scientists should be asked to make themselves available to criticize their research early in the process, even before data collection begins, so that the consistency of method, materials, and analytical approach can be previously peer-reviewed, as was the case with the Registered Report for research on COVID-19 created on the initiative of the Royal Society Open Science magazine [17].

Our survey showed a poor focus on the main aim of the research, i.e., the higher frequency of protocols classified as “other”: this let us suppose that there was no particular attention to the main aim of the study, which could turn into a poor quality of research that probably did not answer primary health needs, especially in light of previous observations that there were mostly observational studies. Many authors have underlined that, in the emergency period, a methodology should be followed to manage studies for new knowledge and to prioritize areas of research [4]. The framework for a good system of health research starts with choosing the research question, with an effort to merge experiences and groups, foster multidisciplinary groups, and coordinate research at an international level to achieve good pooling and avoid overlaps between works [14]. We cannot state for sure that these efforts were lacking in the research under review during the pandemic period, but our data could not highlight good coordination. This was an opportunity for ethical committees to evaluate protocols and try to connect researchers. Since the epidemics of influenza, it was clear that the ethical boards should act fast to avoid losses of knowledge that are of potential benefit for patients, to allow their contribution to have better protocols that should be proportionate to the aim of the research [9].

The main limitation of our study is the poor participation of the Italian ethics committees: we scored an already low overall 20.9% positive response rate, with only a raw 16.5% effectively involved in contributing data. Beyond the mere small size of the sample and the consequent poor representation of the overall Italian situation, the scarce response in participation undermined the aim itself of the present study. We have a low number of studies, mostly observational, from few IECs, but we do not have conclusive information of quality review deterioration under the climate of urgency generated by the COVID-19 pandemic in Italy, nor on quality of protocols. We understood that some committees that did not respond could have been overwhelmed by the amount of work caused by the pandemic; nevertheless, in such a challenging historical period, from both a scientific and political perspective, the need for clear and reliable information is crucial. Our results, such as the lower risk of approving pediatric research and the finding of equally appropriate methodology in both experimental and observational studies, indirectly let us deduce that IECs have faced an effort in holding their main mission, which is the interests of patients for safety and future health benefit.

## 5. Conclusions

The pursuit of quality instead of quantity of information has been a problem during the pandemic, resulting in multiple cases of retractions and withdrawals of scientific papers [8]. We believe that science must be clear and open in its methodology because the general public needs to trust and rely on its methods in order to overcome the challenge of COVID-19 [18,19]. It is our hope that future studies on study quality assessment experience broader participation in order to achieve these fundamental common goals.

## Figures and Tables

**Figure 1 ijerph-19-11673-f001:**
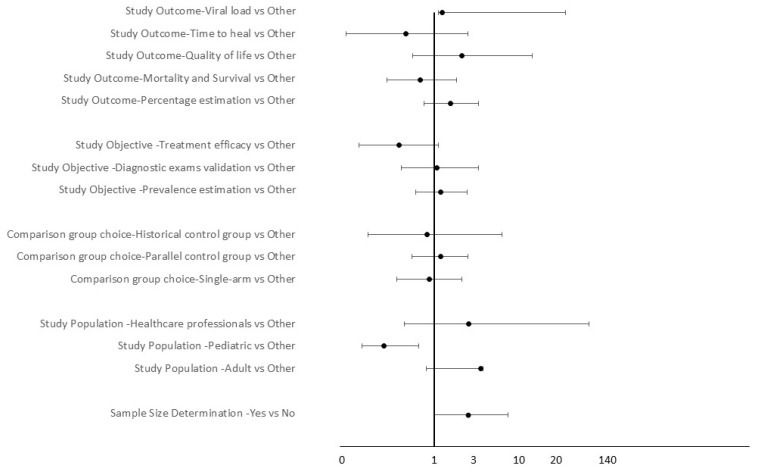
Forest plot showing the odds ratio of approval, and its 95% confidence interval, for each characteristic of the study protocols. When the error bars cross the vertical line, it can be concluded that there is not a statistically significant relation between decision to approve and the characteristic on the left side of the graph.

**Table 1 ijerph-19-11673-t001:** Italian ethics committees participation to the study.

	N.	%
Italian Independent Ethics Committees	91	100.0
Ethics Committees Participation Answers		
No answer to survey	64	70.3
Yes	15	16.5
Yes (No COVID-19 studies)	4	4.4
Not reachable	8	8.8
Study applications	184	100.0
Ethics Committees Contribution by Region		
Lombardia	82	44.6
Liguria	32	17.4
Lazio	17	9.2
Puglia	17	9.2
Sardegna	14	7.6
Campania	11	6.0
Sicilia	11	6.0

**Table 2 ijerph-19-11673-t002:** Study applications’ approval rate and quality of statistical methodology of the studies.

	N.	%
Study applications	184	100.0
Committees Decision		
Approved	127	69.0
Referred for modification	34	18.5
Rejected	6	3.3
Not reported	17	9.2
Statistical Analysis Description		
Yes—appropriate	135	73.4
Yes—not appropriate	8	4.3
No	41	22.3
Sample Size Determination		
No	128	69.6
Yes—appropriate	46	25.0
Yes—not appropriate	9	4.9
Not reported	1	0.5

**Table 3 ijerph-19-11673-t003:** Characteristics of study applications evaluated.

	N.	%
Study applications	184	100.0
Committees decision		
Prospective	71	38.6
Retrospective	46	25.0
Cross-sectional	23	12.5
Experimental	19	10.3
Retrospective–prospective	9	4.9
Diagnostic	9	4.9
Descriptive	6	3.3
Not reported	1	0.5
Study Population		
Adult	145	78.8
Pediatric	22	12.0
Healthcare professional	12	6.5
Other population type	4	2.2
Not reported	1	0.5
Number of centers involved		
Monocentric	112	60.9
Multicentric	72	39.1
Groups creation methods		
Inclusions criteria	114	62.0
Researcher decision	12	6.5
Local feasibility	11	6.0
Other	8	4.3
Randomization	5	2.7
Not reported	34	18.5
Comparison group choice		
Single arm	139	75.5
Parallel control group	32	17.4
Historical control group	9	4.9
Not reported	4	2.2
Study objective		
Other	79	42.9
Prevalence estimation	63	34.2
Diagnostic exams validation	23	12.5
Treatment efficacy	18	9.8
Not reported	1	0.5
Study outcome		
Percentage estimation	65	35.3
Other	53	28.8
Mortality and survival	25	13.6
Quality of life	19	10.3
Time to heal	7	3.8
Therapeutic modifications	6	3.3
Viral load	5	2.7
Not reported	4	2.2

## Data Availability

Restrictions apply to the availability of these data.

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
