# Peer review of "A Survey on Methodological Issues of Clinical Research Studies Reviewed by Independent Ethic Committees during the COVID-19 Pandemic in Italy"

_ijerph, 2022, doi:10.3390/ijerph191811673_

Round 1
Reviewer 1 Report
Getting high quality, ethically grounded research conducted and disseminated is always important, but particularly important during a pandemic, so I agree with the premise of this article. I have a few suggestions for potential improvement.
1. I think some of the most interesting results are in one short paragraph on page 4. This is where some bivariate analyses are discussed. I think this section should be greatly expanded with more discussion on these results and other potential bivariate relationships explored. It is possible the authors have already done this and no significant findings resulted, but in an exploratory paper like this, that would be of interest too.
2. Did the authors explore any differences in the ethics committees that responded and those that didn't on any variables that are available?
3. Would it be possible to share the collected data in an appendix? It would be nice for this paper to inspire other similar works and combining these results with other countries in a meta-analysis might be useful as well.
4. There are several places where the grammar could be improved.
i. In one place "Corona Virus" is used, in another "coronavirus".
ii. On line 14 of page 2, "ensuring solidity and methodology" does not really make sense.
iii. On line 14 of page 6, "because were pressed" should probably be "because they were pressed".
iv. The acronym EBM is used without explanation. Evidence Based Medicine?
That is clearly not an exhaustive list, just a few to illustrate the types of corrections that could be made with a thorough re-reading.
Author Response
Dear reviewer,
thanks for the observations, we have revised the manuscript with the hope to have a better product.
- I think some of the most interesting results are in one short paragraph on page 4. This is where some bivariate analyses are discussed. I think this section should be greatly expanded with more discussion on these results and other potential bivariate relationshipS explored. It is possible the authors have already done this and no-significant findings resulted, but in an exploratory paper like this, that would be of interest too.
R-> Thank you for the comment, we have added the analysis of relation between statistical methodology and type of study (observational vs experimental), sample size determination and type of study, and described as better in the text. Together with the OR of pediatric population study we have added a figure with the OR and its 95% to evaluate risk of approval. So the sentence “approval […] were influenced by any of the variables collected” is completely described.
- Did the authors explore any differences in the ethics committees that responded and those that didn't on any variables that are available?
R-> Thanks for this remark, we have not analysed the differences among Ethical Committees, most of all because they are all regulated under a national rule, so number and qualification of member of the committee have to be equal and homogeneus on the whole nation. The Decree of Ministry of Health in 2006 established that in a Ethical Committee should be one biostatistician, among other members (at least 12 in total counting bioethicist, clinicians, experts in pharmacology, etc…).
Another variable to compare committees could be the volume of studies by year, anyway each region in Italy has allowed a number of committes proportional to the numbers of studies that should be evaluated, therefore we concluded that there will be an homogeneity in numbers of study protocols examined too.
- Would it be possible to share the collected data in an appendix? It would be nice for this paper to inspire other similar works and combining these results with other countries in a meta-analysis might be useful as well.
R-> Thanks for this request and proposal, unfortunately we haven’t the consent of the committees to share data; we have had some difficulties to receive answers to our questionnaire because many committees consider as not appropriate to give us informations on protocols, even if anonimyzed respect to principal investigator, sponsor, title and other carachteristic that helps to identify research protocols. We are open to collaboration giving summarized data as possible.
- There are several places where the grammar could be improved.
- In one place "Corona Virus" is used, in another "coronavirus".
- On line 14 of page 2, "ensuring solidity and methodology" does notreally make sense.
iii. On line 14 of page 6, "because were pressed" should probably be"because they were pressed".
- The acronym EBM is used without explanation. Evidence BasedMedicine?
R-> Thanks for these suggestions, we have revised the whole paper, with particular attention to what is here highlighted.

Reviewer 2 Report
The study is relevant and current. The method is appropriate, although the results have not been complete, as the authors comment in the article. The little response from the committees makes the conclusions less robust. The article can be published without modifications
1. The study addresses an aspect that is difficult to investigate, since it assesses the work of ethics committees in times of pandemic, based on the data offered by the committees themselves, which generates a certain load of subjectivity. 2. Without a doubt, it would have been ideal to obtain greater participation, although it would force each committee to be questioned separately. The results, therefore, are far from complete and this weakens the study. Even so, it is a first approximation to the problem, which invites further investigation. 3. Table 2 indicates that the number of studies in which a sample size determination is made is 55, but further down, under the same heading (Sample Size Determination), it is stated that they are 46. This should be clarified. 4. Perhaps the claims could have been supported by a more extensive bibliography. Along these same lines, the Conclusion could have been more extensive and defined, taking into account everything provided in the Discussion.
Reviewer 3 Report
The study “Ethics Committees Reviews of Clinical Research Studies. Applications in Italy during the Covid-19 Pandemic” aimed to assess the impact of Covid-19 on clinical research quality submitted to Italian Ethics Committees in the period between April and July 2020.
Major comments:
1. The title gives the reader the impression that the manuscript presents a review of the works of Ethics Committees in Italy during the Covid-19 pandemic, but in fact, an attempt is made to assess the quality of the reviewed protocols. The title should change accordingly to reflect the aim of the study. For instance, the title could change to “Quality of clinical research protocols reviewed by Ethics Committees in Italy during the Covid-19 pandemic”.
2. My main criticism on the study is whether the data that were available to the researchers were adequate and sufficient to draw robust conclusions on the quality of the reviewed protocols by the Ethics Committees. Although power calculation and sample size are critical parameters in research design, yet these statistical factors alone are not sufficient to judge the quality of research and the quality of the followed protocols, particularly because the authors did not have access to the protocols per se (and of course they could not have access due to confidentiality). Focusing on these aspects only seems as a bias and weakens the study and its conclusions.
The ethical review process by Ethics Committees considers also some fundamental aspects of research protocols such as risks and benefits for the participants and completeness and comprehensibility of information provided to participants (as the authors point out in the Introduction). These factors were not taken into consideration in the present study to judge quality of reviewed research, since good research is also ethical research.
As mentioned in the Introduction, another issue which is particularly relevant for the review of research protocols by Ethics Committees during the COVID-19 pandemic is the time needed to review a protocol. Again this factor (fast track or not) was not taken into consideration in the present study.
3. In the discussion the authors argue that “Beyond the mere downsizing of the sample numerosity and the consequent poor representativity of the overall Italian situation, the scarce response in participation, against the Ethics Committees’ self-interest, undermined the aim itself of the present study, making us unable to disprove the suspicion of quality review deterioration under the climate of urgency generated by Covid-19 pandemic in Italy.” is an absolutely imprudent conclusion which is not justified. The fact that Ethics Committees did not respond to the survey could be due to various reasons and cannot provide any information regarding the quality of the reviewed protocols, and points towards an initial bias in the study hypothesis.
Minor comments:
1. The English language needs to be reviewed again. For instance (only two examples are given):
“Those were the key point that moved to valuable results and appreciation of procedures” should be “These were the key points that moved to valuable results and appreciation of procedures”.
“Since the epidemics of influenza, it was clear that the ethical board should act fast to avoid losses of knowledge that are potential benefit for patients, to give their contribution to have better protocols and they should be proportionate to the aim of the research” does not make sense, the sentence needs to be revised.
2. The Introduction and Discussion need to be broken down in paragraphs to be readable and comprehensible.
Round 2
Reviewer 1 Report
The authors addressed my concerns.
Reviewer 3 Report
The authors have revised the manuscript and adequately addressed the issues highlited after the 1st review. I recommend acceptance of the manuscript in its current form, but after necessary proofreading to correct for some remaining english language errors.